# Crosstalk Among Gut Microbiota, Microbial Metabolites, and Inflammatory Cytokines: Current Understanding and Future Directions

**DOI:** 10.3390/foods14223836

**Published:** 2025-11-09

**Authors:** Guanglei Wu, Ran Wang, Yicheng Wang, Siyuan Sun, Juan Chen, Qi Zhang

**Affiliations:** 1College of Food Science and Nutritional Engineering, China Agricultural University, Beijing 100083, China; wuguanglei@cau.edu.cn; 2Department of Nutrition and Health, China Agricultural University, Beijing 100190, China; 3Key Laboratory of Functional Dairy, Co-Constructed by Ministry of Education and Beijing Government, China Agricultural University, Beijing 100091, China

**Keywords:** gut microbiota, microbial metabolites, inflammatory cytokines, antibiotics, probiotics

## Abstract

The interaction between the gut microbiota and the host immune system is pivotal in maintaining health or driving disease pathogenesis. The gut microbiota directly or indirectly modulates immune cells activation and inflammatory cytokines secretion through microbial metabolites, including short-chain fatty acids (SCFAs), tryptophan metabolites, bile acids, and polyamines. Conversely, the immune system regulates microbial community composition by maintaining the integrity of the epithelial barrier. In addition, antibiotics and probiotics can further regulate the inflammatory response by altering gut microbiota structure and microbial metabolites levels. This review systematically examines the bidirectional regulatory mechanisms among the gut microbiota, microbial metabolites, and inflammatory cytokines, and explores the impact of antibiotics and probiotics on this interaction network. These insights provide new targets for immune-related diseases.

## 1. Introduction

The gut microbiota, often referred to as the human body’s “second brain,” is a diverse and dynamic microbial community colonizing the gastrointestinal tract [1,2]. Its composition and functional capacity are critically important to human health. Under physiological conditions, the gut microbiota contributes to host well-being by metabolizing dietary components, synthesizing essential nutrients, resisting pathogen colonization, and modulating immune responses [3,4,5]. Much like other organs, the composition and diversity of the gut microbiota are closely linked to host physiological functions such as nutrient metabolism, biosynthesis, barrier protection, and immune homeostasis [6]. According to the International Code of Nomenclature of Prokaryotes (ICNP), the gut microbiota is classified into various taxonomic ranks [7]. Major phyla include *Bacillota*, *Bacteroidota*, *Actinomycetota*, *Fusobacteriota*, *Pseudomonadota*, and *Verrucomicrobiota*. The composition of the gut microbiota is not static; rather, its stability is co-regulated by both host genetics and external environmental factors. Among these external influences, dietary patterns (e.g., high-fiber or high-fat diets), antibiotic use, lifestyle, stress, and disease states can significantly influence gut microbial diversity [8,9].

The gut microbiota plays a crucial role in maintaining immune defense functions. Under normal physiological conditions, it exhibits strong ecological resilience, enabling it to restore its original composition through self-regulation mechanisms after minor disturbances [10]. However, long-term antibiotic abuse or changes in dietary patterns can lead to irreversible alterations in the gut microbiota structure, a condition known as dysbiosis [11]. Dysbiosis is primarily characterized by reduced microbial diversity, a decline in beneficial bacteria, and an increase in potentially pathogenic bacteria [12,13]. This imbalance can trigger systemic low-grade inflammation through gut microbiota-immune interactions, manifesting as weakened anti-inflammatory capacity and enhanced pro-inflammatory responses [14]. Consequently, dysbiosis drives the initiation and progression of various chronic diseases, including inflammatory bowel disease (IBD), metabolic syndrome, autoimmune disorders, and neurodegenerative diseases. Furthermore, structural disruption of the gut microbiota alters its metabolic output, reducing the synthesis of beneficial metabolites [15] (e.g., short-chain fatty acids [SCFAs] like butyrate and propionate) while increasing production of harmful metabolites (e.g., lipopolysaccharides, LPS) [16]. These changes directly impair intestinal epithelial barrier function, induce “leaky gut,” and disrupt local and systemic immune responses [17].

Conversely, the immune system also regulates microbial community composition by maintaining epithelial barrier function. This immunomodulatory is mediated through three major synergistic mechanisms: First, inflammatory cytokines (e.g., TNF-α, IL-4, and IL-13) produced by the immune system modulate the expression of mucins (e.g., MUC2) in goblet cells [18,19]. These mucins, upon glycosylation, form glycan structures that serve as key carbon sources for certain gut bacteria (e.g., *Akkermansia muciniphila* and *Bacteroides* spp.) [20]. Changes in the availability of mucin glycans directly affect the abundance of glycan-dependent bacteria, thereby reshaping the overall composition of the gut microbiota [12]. Second, immune cells secrete IgA antibodies that precisely coat and suppress mucosa-associated bacteria, particularly potential pathogens, limiting their adhesion, colonization, and translocation. This process helps maintain the spatial distribution and ecological balance of the microbiota [21]. Third, immune signaling enhances the expression of epithelial tight junction proteins and promotes Paneth cell secretion of antimicrobial peptides (e.g., defensins), further strengthening physical and chemical barrier functions [22]. These actions shape the microbial microenvironment and enable deep regulation of the microbiota’s structure and function [23]. Overall, the immune system employs these multiple mechanisms to synergistically maintain intestinal epithelial barrier integrity, directly or indirectly regulating the composition of the gut microbiota.

In summary, there exists a close and complex bidirectional regulatory relationship among the gut microbiota, its metabolites, and the immune system’s inflammatory cytokines [24]. On one hand, the gut microbiota and its metabolites directly regulate immune cell functions and inflammatory cytokine expression [25,26]. On the other hand, the immune system feedback-modulates microbial structure and ecological stability through cytokines and immune mediators [27]. This intricate crosstalk is central to maintaining gut homeostasis. Therefore, this review discusses the close interactions among the gut microbiota, microbial metabolites (e.g., SCFAs, bile acids, and tryptophan metabolites), and inflammatory cytokines, providing strategies for developing gut microbiota-based interventions (e.g., probiotics), with the aim of enhancing host immune function.

## 2. Gut Microbiota Modulate the Release of Inflammatory Cytokines

The intestinal immune system is primarily composed of the gut mucosa, immune cells, and associated metabolites [28]. The gut microbiota interacts directly or indirectly with host immune cells through microbial metabolites, leading to the production of inflammatory cytokines [29]. These cytokines serve as key signaling molecules that stimulate B cells, T cells, or lymphocytes to release specific antibodies in response to pathogenic insults, thereby triggering immune responses to maintain intestinal homeostasis [30,31]. In the context of intestinal immunity and inflammatory responses, cytokines can be broadly classified into two functional categories [32,33]: pro-inflammatory and anti-inflammatory. Typical pro-inflammatory cytokines, such as IL-6, IL-8, IL-12, TNF-α, and IFN-γ, are primarily produced by immune cells (e.g., macrophages and T cells) and epithelial cells [34]. Their functions include activating immune responses, mediating chemotaxis, and facilitating pathogen clearance. In contrast, anti-inflammatory cytokines such as IL-10 and TGF-β are mainly secreted by regulatory T cells and dendritic cells [35], and function to suppress excessive inflammation, promote tissue repair, and maintain immune homeostasis [36]. IL-6, a pleiotropic cytokine, exhibits both pro- and anti-inflammatory properties, with its specific role depending on the gut microenvironment and relevant receptor signaling [37]. A systematic summary of the cellular origins and physiological functions of key intestinal inflammatory cytokines will help further elucidate the interaction mechanisms between gut microbiota and inflammatory cytokines (Table 1).

The gut microbiota serves as a critical regulatory hub by modulating the release of inflammatory cytokines, thereby playing a central role in the transition between systemic homeostasis and disease pathogenesis [47,48]. Accumulating evidence has established gut dysbiosis as a key factor disrupting the inflammatory cytokine network. Accompanied by damage to the intestinal mucosal barrier, dysbiosis disrupts the balance of inflammatory cytokines [14]. Dysregulation of inflammatory mediators—characterized by excessive pro-inflammatory cytokine production and/or impaired anti-inflammatory mechanisms—directly promotes the initiation and progression of various chronic diseases, including autoimmune conditions, metabolic syndrome, and neurodegenerative disorders [49] (Table 2). For instance, in diabetic patients, decreased abundance of *Roseburia*, *Faecalibacterium*, *Bifidobacterium*, and *Akkermansia* contributes to intestinal barrier dysfunction [50,51,52,53,54]. This permits bacteria or bacterial products such as lipopolysaccharide (LPS) to enter the circulation. Translocated LPS binds to TLR4 receptors on adipocytes, activating NF-κB signaling and stimulating the production of pro-inflammatory cytokines, including TNF-α and IL-6 [16]. These cytokines inhibit insulin signaling pathways, induce insulin resistance, and propagate an inflammatory cascade [55]. In sepsis patients, elevated levels of *Enterococcus*, *Streptococcus*, and *Staphylococcus* correlate with increased systemic concentrations of cytokines such as IL-6 and IL-10 [56]. Similarly, Chen et al. observed that adolescents with depression exhibit reduced abundance of *Bifidobacterium*, *Escherichia*, *Lactobacillus*, and *Bacteroides*, along with heightened levels of pro-inflammatory cytokines, including TNF-α, IL-1β, and IL-6 [57]. Moreover, increased gut epithelial permeability facilitates the translocation of other immunogenic substances—such as intact proteins, gluten, and food antigens—into the bloodstream. Upon systemic dissemination, these antigens trigger an immune response culminating in the production of autoantibodies, exaggerated immune activation, and the development of autoimmune diseases, including celiac disease, systemic lupus erythematosus (SLE), and autoimmune hepatitis [58,59].

Research on the gut microbiota and its metabolites offers novel perspectives for innovating strategies in disease diagnosis and treatment. For instance, in the field of inflammatory bowel disease (IBD), the gut microbiota has emerged as a critical therapeutic target [70]. Modulation of the gut microbiota is also considered a promising treatment approach for hypertension [71]. In the management of gastrointestinal cancers and infectious diseases, microbial metabolites demonstrate considerable potential by engaging multi-dimensional regulatory mechanisms [72]. Targeting these metabolites allows for effective modulation of host immune responses, thereby slowing disease progression. Inflammatory cytokines act as key messengers in cardiometabolic diseases and various chronic conditions, playing a central role in disease pathogenesis. Essentially, regulating the gut microbiota and its metabolites can indirectly modulate cytokine levels, offering innovative avenues for preventing and treating these diseases [73].

Research confirmed that the gut microbiota modulates the release of inflammatory cytokines by either directly colonizing or indirectly adhering to epithelial cells (Figure 1). Firstly, certain gut microorganisms, such as invasive *Klebsiella* species, can directly colonize intestinal epithelial cells. These organisms are phagocytosed by dendritic cells (DCs), stimulating the release of IL-6, TNF, and IL-12 [74]. Secondly, the surface components of gut microbes, including extracellular polysaccharides (EPS) and pili, enhance their adhesion to intestinal epithelial cells [75]. For instance, *Bifidobacterium* with a thicker EPS layer not only directly regulates the secretion of IL-10 and TNF-α by macrophages but also modulates the secretion of IL-6 by DCs via the Toll-like receptor 2 (TLR2) receptor [76]. Additionally, *Bifidobacterium* with pili enhances the adhesion to intestinal epithelial cells and regulates the secretion of TNF-α by macrophages compared to no pili control strains [77]. Thirdly, segmented filamentous bacteria (SFB) stimulate the secretion of TGF-β, IL-12, IL-23, and IL-17 by CD11c+ DC cells, mediated by the release of serum amyloid A (SAA) from epithelial cells. Fourthly, specific gut microbes, such as *Bacteroides* species or *Clostridium clusters* IV and XIVa, can directly regulate CD103+ DC cells through goblet cells, promoting the secretion of TGF-β and IL-10 [78]. Furthermore, *Firmicutes* with substrate-binding protein (SBP) and *Bacteroidetes* with tetratricopeptide repeat lipoprotein (TPRL) can selectively recognize T-cell receptors (TCRs), thereby regulating the secretion of TNF-α and IFN-γ [79]. In summary, the gut microbiota can directly or indirectly regulate the release of inflammatory cytokines, thereby influencing host immune function.

## 3. Microbial Metabolites Modulate the Release of Inflammatory Cytokines

The gut microbiota can directly or indirectly modulate host immune responses through microbial metabolites, thereby contributing to the maintenance of immune homeostasis. Recent studies have revealed that the majority of systemic metabolites are gut-derived and influence immune reactivity by regulating the differentiation, proliferation, and apoptosis of immune cells [80]. In parallel, advances in both non-targeted and targeted metabolomics technologies have enabled the identification and detection of an increasing number of small-molecule metabolites associated with the gut microbiota. These metabolites can be categorized into three main groups (Figure 1). (1) Metabolites produced from dietary precursors via indispensable gut microbial biotransformation (e.g., short-chain fatty acids and tryptophan metabolites; not obtainable directly from diet) [81]. (2) Metabolites from host-microbial co-metabolism: host-synthesized molecules that require structural modification by gut microbial enzymes to alter their biological functions, as exemplified by bile acids [82]. (3) Microbially de novo synthesized metabolites: metabolites directly produced and released by gut microbiota (e.g., LPS; polyamines, primarily derived from microbes in the gut) [83]. Furthermore, it is now widely acknowledged among experts that these metabolites influence host immune function by modulating inflammatory factors through interactions with immune cells.

### 3.1. Short-Chain Fatty Acids (SCFAs)

Short-chain fatty acids (SCFAs) are byproducts of the gut microbiota’s metabolism of undigested carbohydrates. Among these, acetate, propionate, and butyrate are the most abundant SCFAs [84]. These organic acids play significant roles in immune regulation through several mechanisms. Firstly, SCFAs serve as energy sources for intestinal epithelial cells and can directly bind to G protein-coupled receptors (GPCRs) such as GPR43, GPR41, and GPR109A, thereby triggering the release of the pro-inflammatory cytokine IL-18 [85]. Secondly, butyrate, specifically, can induce the differentiation of Foxp3+ regulatory T cells (Tregs) by modulating histone deacetylase (HDAC) activity, thereby influencing IL-18 expression [86]. Thirdly, butyrate binds to GPCRs on DC cells, inducing the release of IL-10 [87]. Butyrate also directly influences the plasticity of type 3 innate lymphoid cells (ILC3s), enhancing their production of the barrier-protective cytokine IL-22 [88].

### 3.2. Tryptophan Metabolites

The gut microbiota, including *Escherichia coli*, *Proteus vulgaris*, *Clostridium*, and *Ruminococcus*, can metabolize tryptophan into various metabolites such as indole, indole ethanol (IE), indole propionic acid (IPA), indole lactic acid (ILA), indole acetic acid (IAA), indole aldehyde (IAld), indole acrylic acid (IA), skatole, and tryptamine. This process occurs through the catalytic enzymes involved in tryptophan metabolism or via decarboxylation reactions [89,90]. Tryptophan metabolites primarily regulate the levels of IL-10 and IL-22 by acting on B cells through the aryl hydrocarbon receptor (AhR) [91]. For example, *Bacteroides* and *Taylorella* species facilitate the production of tryptophan-derived metabolites, such as IAA and IPA, which act as agonists of AhR [92,93,94]. Upregulation of AhR promotes the differentiation of naïve T cells into Th17 and regulatory T (Treg) cells, thereby upregulating the expression of the anti-inflammatory cytokine IL-10 while suppressing the production of pro-inflammatory cytokines such as IFN-γ and IL-17. This mechanism has been demonstrated to exert anti-inflammatory effects in dextran sulfate sodium (DSS)-induced colitis models [95]. In rodent models of diabetes-associated depression, activation of indoleamine 2,3-dioxygenase (IDO)—a key enzyme in tryptophan metabolism—results in decreased hippocampal serotonin (5-HT) levels and elevated pro-inflammatory cytokines such as TNF-α, IL-1β, and IL-6 [96]. Nevertheless, current research on the regulation of inflammatory cytokines by these metabolites remains limited, and other potential signaling pathways involved warrant further exploration.

### 3.3. Bile Acids (BAs)

BAs encompass two categories: primary and secondary bile acids. Approximately 5% of primary BAs are transformed into secondary BAs through metabolism by the gut microbiota, notably by *Clostridium scindens* within the Firmicutes phylum, via the action of 7α-hydroxylase [97]. Several key mechanisms underlie how BAs modulate inflammatory cytokines. For instance, secondary BAs like lithocholic acid (LCA) can directly stimulate Th17 and Treg cells, thereby influencing the levels of inflammatory cytokines [98]. Additionally, BAs can bind to VDR receptors on Treg cells, stimulating the release of inflammatory cytokines [81]. BAs can also bind to the FXR receptor on DC cells or the TGR5 receptor on macrophages, further regulating the concentration of inflammatory cytokines [99].

### 3.4. Polyamines

Polyamines mainly include spermine, spermidine, and putrescine. Gut microbiota, such as *Bacteroides* and *Clostridium*, are major producers of polyamines [100]. Carriche G. M. et al. confirmed that dietary spermidine supplementation can promote the steady-state differentiation of Treg cells in the gut, but its regulatory effect on inflammatory cytokines has not been studied in more detail [101].

### 3.5. Microbe-Associated Molecular Patterns (MAMPs)

Lipopolysaccharide (LPS) functions as a crucial priming agent for the activation of the NLRP3 inflammasome. It initiates the activation of caspase-1, which subsequently catalyzes the maturation and release of the pro-inflammatory cytokines IL-1β and IL-18, thereby coordinating a targeted innate immune response [102]. This pathway represents a fundamental aspect of host defense and is also significantly involved in the pathogenesis of various inflammatory diseases, including Alzheimer’s disease, sepsis, and autoimmune disorders [103,104]. Similarly, other microbe-associated molecular patterns contribute to immune activation through distinct pathways. Lipoteichoic acid, a key component of the Gram-positive bacterial cell wall, triggers inflammation via the TLR2 signaling pathway [105]. Correspondingly, ubiquitous peptidoglycan fragments are detected by the intracellular pattern recognition receptors NOD1 and NOD2 [106]. This recognition subsequently initiates the activation of the NF-κB pathway and drives inflammatory responses [107]. Furthermore, bacterial extracellular vesicles (BEVs)—nanoscale particles released by bacteria—function as central hubs for immune coordination. These vesicles synergistically activate both TLR and NOD signaling pathways while also serving as potent activators of the NLRP3 inflammasome, significantly enhancing the maturation and release of IL-1β [108,109,110].

## 4. Inflammatory Cytokines Remodel the Gut Microbiota Composition

The composition of gut microbiota can be either directly or indirectly regulated by host immune cells, such as macrophages, T cells, and B cells, through the action of inflammatory cytokines (Figure 2). Firstly, the composition of the gut microbiota can be regulated by the secretion of two antimicrobial peptides (AMPs), α-defensin or β-defensin, by macrophages, natural killer T cells (NKT), and B cells [111]. For instance, α-defensins can regulate the overgrowth of *Clostridium difficile* [112]. While β-defensin-hBD-1 is only active against Gram-positive bacteria, β-defensin HBD-2, 3, and 4 exhibit antibacterial action against *Escherichia coli* (*E. coli*), *Pseudomonas aeruginosa*, *Staphylococcus aureus*, and *Streptococcus pyogenes*. Secondly, TL1A and IL-18 release can be regulated by TCRγδ(+), CD8(+) T cells, and muco-associated immutable T cells (MAIT), which in turn can alter the ratio of *Bacillota*/*Bacteroidetes*, as well as the contents of *Lactobacillus* and *E. coli* [113,114]. Thirdly, NKT can selectively identify lipid antigens presented by CD1d on DC cells and intestinal epithelial cells. Additionally, NKT can control the makeup of symbiotic microbes such as *Lactobacillus gasseri*, *Staphylococcus aureus*, and *E. coli* [115]. Lastly, polymerized immunoglobulin receptors (pIgR) allow IgA antibodies made by B cells and follicular helper T (Tfh) cells to enter the intestinal lumen and control the composition of the gut microbiota [116].

## 5. The Evolutionary Pressure by Antibiotics on the Communication Network

Antibiotics have long been considered effective agents for treating bacterial infections; however, their widespread and often indiscriminate use can exert long-lasting impacts on the gut microbiota through drug-specific mechanisms, thereby increasing the host’s susceptibility to various diseases [117,118]. Studies indicate that both short-term and long-term antibiotic treatments can disrupt the gut microbiota, leading to reduced microbial diversity, loss of ecological balance, and overgrowth of pathogenic bacteria such as *Clostridium difficile* [119]. For instance, the use of β-lactams, glycopeptides, and macrolides has been associated with decreased abundance of beneficial bacteria including *Bifidobacterium* and *Lactobacillus* [120]. Research in animal models has revealed that early-life administration of antibiotics like tylosin and amoxicillin reduces the abundance of taxa such as *Muribaculaceae*, *S24-7*, *α-proteobacteria*, and *δ-proteobacteria*. These alterations increase susceptibility to pathogens and exacerbate the severity of diseases in adulthood, including diarrhea, hemorrhagic colitis, Crohn’s disease, and colorectal tumors [121]. Clinical studies further confirm that even a brief four-day course of combined meropenem, gentamicin, and vancomycin can induce a substantial expansion of *Enterobacteriaceae*, *Enterococcus faecalis*, and *Fusobacterium nucleatum*, alongside a concurrent reduction in *Bifidobacterium* and butyrate-producing bacterial populations [122]. Moreover, exposure to antibiotics during early life, especially in infancy, may cause long-term disruptions to the composition of the gut microbiota. Evidence from a Finnish pediatric cohort revealed that early macrolide use led to decreased abundance of *Actinobacteria* and increased levels of *Bacteroidetes* and *Proteobacteria*. These microbial alterations are associated with elevated risks of asthma and obesity in later childhood [123]. In addition to the direct adverse outcomes of microbiota disruption and disease induction, antibiotic administration poses a more profound and enduring threat: the emergence and dissemination of antibiotic resistance [124]. The powerful selective pressure exerted by these drugs promotes the rapid enrichment and ecological dominance of resistant bacterial strains within the host [125,126,127]. These resistant strains can also rapidly propagate resistance genes across different species via horizontal gene transfer mechanisms such as conjugation, transduction, and transformation [128]. This enables the widespread dissemination of resistance traits throughout microbial communities, further compromising treatment efficacy and facilitating the rise in multidrug-resistant infections [117].

Beyond directly altering the microbial composition, antibiotics also significantly impact the levels of microbial metabolites. Zhang et al. [129] demonstrated that after three weeks of vancomycin treatment, the fecal contents of mice showed significant reductions in short- and long-chain fatty acids, bile acids, L-arginine, L-tyrosine, and phosphatidylcholines. Further research [130] revealed that treatment with enrofloxacin, vancomycin, and polymyxin B upregulates the expression of colonic cytokine-related genes and significantly disrupts the biosynthesis pathways of valine, leucine, and isoleucine. Moreover, the combined administration of vancomycin with ciprofloxacin–metronidazole markedly reduced the concentrations of alanine, branched-chain amino acids, and aromatic amino acids in the colonic contents of female mice [131].

Antibiotic-induced disturbances in gut microbiota and metabolite profiles are often accompanied by shifts in inflammatory cytokine production. Notably, the expression levels of inflammatory cytokines such as IL-17A, IL-22, IL-1β, and IL-12 are affected by antibiotics [132,133]. Wang et al. [134] identified positive correlations between *Enterococcus* and *Klebsiella* genera and pro-inflammatory mediators such as TNF-α, IL-12, and IL-1β. Kathryn A Knoop et al. [135] found that levels of the inflammatory cytokines CXCL 1, IL-17, and IFN γ were significantly elevated after sustained antibiotic use. Collectively, these findings highlight a tightly intertwined relationship among antibiotic usage, gut microbial community structure, metabolite dynamics, and inflammatory immune responses.

## 6. Foods Intervene in the Immune-Inflammatory Response by Modulating Inflammatory Cytokines

### 6.1. Probiotics

Probiotics, defined by the World Health Organization (WHO) and the Food and Agriculture Organization (FAO) of the United Nations as “live microorganisms which when administered in adequate amounts confer a health benefit on the host,” [136] represent a low-cost and well-tolerated intervention strategy for managing inflammatory conditions. In recent years, probiotic strains, particularly *Bifidobacterium* and *Lactobacillus*, have emerged as promising microbiota-targeting therapies [137,138]. They are now widely used as adjunctive or alternative treatments for a range of immune-related inflammatory diseases [139].

Probiotics exert their beneficial effects through multiple synergistic mechanisms to maintain intestinal homeostasis. Firstly, as live microorganisms, they are capable of colonizing and proliferating within the host intestine, thereby increasing the abundance of beneficial bacteria and establishing ecological dominance [140,141]. Secondly, through competitive exclusion, probiotics compete with pathogens for nutrients and adhesion sites. This competition facilitates the expansion of commensal bacteria while suppressing the growth of harmful microbes [142]. Thirdly, probiotics produce metabolites such as SCFAs, which lower the luminal pH and create an unfavorable environment for enteropathogens [143,144]. Certain strains also secrete antimicrobial substances—including bacteriocins, lactic acid, and defensins—that directly inhibit pathogen colonization [145]. In terms of immunomodulation, probiotics fine-tune host immunity via direct interactions with immune cells [146]. Strains from different genera can engage with dendritic cells (DCs), prompting the differentiation of naïve T cells into various subsets such as Th1, Th2, or regulatory T cells (Tregs) [147]. This polarization, in turn, stimulates the production of specific inflammatory cytokines. For instance, *Lactobacillus* species can directly modulate T-cell responses by enhancing the secretion of cytokines including TGF-β, IL-10, and IL-8, thereby precisely calibrating the intensity and direction of immune activation [148]. Through such multi-layered crosstalk with the immune system, probiotics not only enhance innate and adaptive immunity but also help suppress excessive inflammation and mitigate the development of inflammation-associated disorders.

Regarding the immunomodulatory functions of probiotics within cytokine networks, accumulating evidence indicates that their beneficial effects are largely mediated through a dual regulatory mechanism. This involves the suppression of pro-inflammatory cytokines, including TNF-α, IL-6, and IL-8, coupled with the promotion of anti-inflammatory cytokine secretion in the gut [149]. Sun et al. [150] utilized untargeted metabolomics to investigate the anti-inflammatory properties of a postbiotic produced by *Lactobacillus paracasei* K56—a strain initially isolated from infant gut microbiota. Their study demonstrated that the cell-free supernatant (CFS) of *L. paracasei* K56 significantly reduced TNF-α expression in macrophage RAW 264.7 cells, effectively mitigating cellular inflammation. These anti-inflammatory effects were further validated in both a high-fat diet-induced zebrafish model and a DSS-induced murine model of ulcerative colitis. The anti-inflammatory properties of probiotics and postbiotics have been consistently demonstrated across various animal disease models. The DSS-induced mouse colitis model has demonstrated that *Lactobacillus rhamnosus* GG suppresses inflammation by upregulating IL-10 expression in Ly6C^+^ monocyte [135]. Similarly, *Bifidobacterium infantis* promotes Treg cell differentiation and enhances the secretion of IL-10 and TGF-β1 [151]. Furthermore, *L. rhamnosus* ZFM231 significantly alleviates the pathological progression of colitis by modulating the TGF-β/TGF-α balance and regulating the microbiota structure [152]. This immunomodulatory effect has been further validated in clinical trials. *Bifidobacterium longum* ES1 significantly reduces serum levels of IL-6, IL-8, and TNF-α in patients with irritable bowel syndrome (IBS), while restoring intestinal permeability and barrier function [153]. In the context of immune inflammation, *Lactiplantibacillus plantarum* HM-22 effectively corrects the Th1/Th2 immune imbalance in an α-lactalbumin (α-LA)-induced allergy model by markedly upregulating the expression of anti-inflammatory factors IL-10, IFN-γ, and TGF-β, while inhibiting Immunoglobulin E (IgE) production and the secretion of the Th2-type cytokine IL-4 [154]. A clinical trial involving patients with atopic dermatitis demonstrated that oral administration of *L. plantarum* IS-10506 significantly reduced serum levels of IL-4 and IL-17, while markedly increasing levels of IFN-γ and Foxp3+ [155]. In the field of neurodegenerative diseases, research on Parkinson’s disease (PD) has demonstrated that *L. salivarius* LS01 and *L. acidophilus* significantly reduce levels of TNF-α, IL-6, and IL-17A, while enhancing the expression of IL-4 and IL-10 through the stimulation of peripheral blood mononuclear cells (PBMCs) in PD patients [156]. Additionally, in the MPTP-induced PD mouse model, interventions with *L. plantarum* CRL 2130 and *Streptococcus thermophilus* (CRL 807/808) resulted in decreased serum levels of IL-6 and TNF-α, along with an increase in IL-10, which was significantly associated with the improvement of PD symptoms [157]. A diverse collection of probiotic strains, tested across various experimental systems—including in vitro studies, animal models, and clinical trials—demonstrates therapeutic potential in preventing and treating immune-related diseases via modulation of inflammatory cytokines (Table 3).

Driven by the rapid advancement of gene editing technologies, therapeutic strategies for immune-related diseases continue to expand. In the future, interventional approaches will no longer be limited to conventional probiotics; engineered probiotics are emerging as a more precise and efficient treatment modality [159,160]. As live biotherapeutic products, they demonstrate significant potential for targeting specific diseases. These programmed probiotics can secrete bioactive metabolites such as SCFAs, which precisely interact with disease markers and related effector molecules, thereby modulating specific pathological pathways [161]. Leveraging this mechanism, engineered probiotics have been applied in various fields, including metabolic disorders, behavioral abnormalities, and cancer therapy [162,163]. For instance, Praveschotinunt et al. [164] employed genetic engineering techniques to enable *Escherichia coli Nissle* 1917 (EcN) to express the anti-inflammatory cytokine IL-10. This modified strain not only promoted the expansion of beneficial microbial populations and improved the gut microenvironment but also markedly attenuated inflammatory responses, effectively alleviating symptoms associated with various intestinal diseases [165]. This example underscores the practical value and broad prospects of engineered probiotics in disease treatment.

### 6.2. Other Foods

A diverse range of interventions beyond probiotics—including prebiotics, postbiotics, synbiotics, fecal microbiota transplantation (FMT), and dietary modulation—has shown considerable potential in alleviating inflammatory diseases through the regulation of gut microbiota [166]. Prebiotics, selectively utilized by host microorganisms, promote the growth and colonization of beneficial bacteria via fermentation-derived metabolites such as SCFAs [167]. Research has demonstrated that prebiotics can mitigate periodontitis by downregulating pro-inflammatory cytokines, including TNF-α and IL-1β [168]. Postbiotics, characterized by their well-defined composition, stability, and safety, have exhibited superior efficacy in colitis models. Research in a DSS-induced mouse colitis model demonstrates that postbiotics from *Saccharomyces boulardii* more effectively lower serum TNF-α and IL-6 levels than live bacteria [169]. Complementing this, heat-inactivated *Lactiplantibacillus argentoratensis* BBLB001 boosts colonic mucin and secretory IgA, underscoring the critical function of postbiotics in reinforcing intestinal defense and microbial regulation [170]. Moreover, supernatants from *Lactobacillus acidophilus* and *Lactobacillus casei* suppress the release of the pro-inflammatory cytokine TNF-α and promote that of the anti-inflammatory IL-10 in macrophages, thereby enhancing intestinal barrier integrity and fine-tuning local immunity [171]. Synbiotics, which integrate probiotics and prebiotics in a complementary manner, have demonstrated therapeutic potential in inflammatory bowel disease [172]. In contrast, fecal microbiota transplantation (FMT) restores a healthy gut microbial ecosystem and has proven effective in treating recurrent *Clostridioides difficile* infections and metabolic syndromes [173,174]. Collectively, these integrated microecological strategies not only expand the understanding of gut–immune axis regulation but also facilitate the development of novel dietary interventions based on functional foods, underscoring their promising clinical applicability.

## 7. Conclusions

Numerous studies have been conducted on the relationships among inflammatory cytokines, gut microbiota, and gut metabolites. One possible therapeutic strategy is to control immunity through gut microbiota and its metabolites. Interestingly, a large number of studies on probiotics have confirmed that probiotics can regulate the release of inflammatory cytokines through gut microbiota and metabolites, but different types of probiotics, different doses, and different intervention times have different therapeutic effects. The gut microbiota and the immune system together form a complex network, and their relationship requires further investigation through in vitro or in vivo studies, including animal or clinical studies. Meanwhile, consideration must be given to the diverse immune cell types, the multiple levels of gut microbiota classification, and the varying conditions required for metabolite identification. In summary, further research into the interrelationships among these three aspects is expected to provide a robust foundation for enhancing immunity through gut microbiota modulation.

## Figures and Tables

**Figure 1 foods-14-03836-f001:**
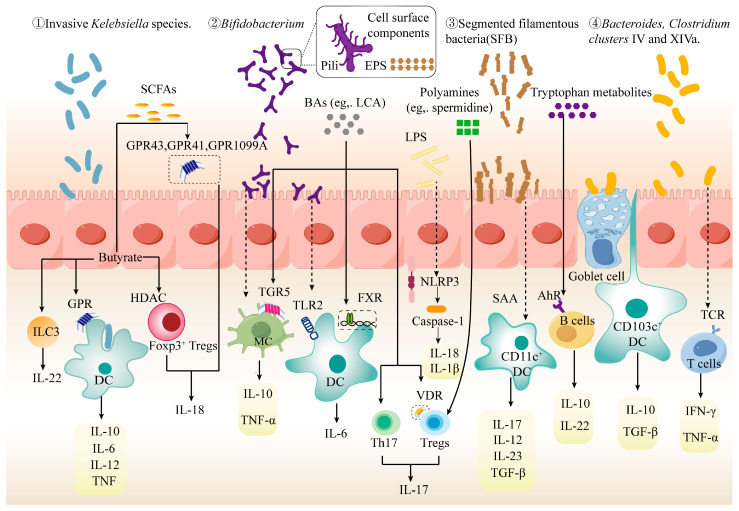
The gut microbiota and metabolites regulate the release of inflammatory cytokines. The gut microbiota regulates inflammatory cytokine release through immune cells mainly by (1) direct adhesion to intestinal epithelial cells (e.g., ① invasive *Klebsiella* species with invasive ability; ② *Bifidobacterium* with extracellular polysaccharides (EPS) and pili); and (2) indirectly through goblet cells. (e.g., ③ segmented filamentous bacteria (SFB) that stimulate serum amyloid A (SAA) release; ④ *Bacteroides* species or *Clostridium clusters* IV and XIVa that pass through goblet cells). Interestingly, gut microbiota metabolites including: short-chain fatty acids (SCFA), tryptophan metabolites, bile acids (BAs), lipopolysaccharide (LPS), polyamines, etc., all directly or indirectly modulate the relevant receptors, which further stimulate the release of inflammatory cytokines from immune cells such as B cells, T cells (Tregs), dendritic cells (DCs).

**Figure 2 foods-14-03836-f002:**
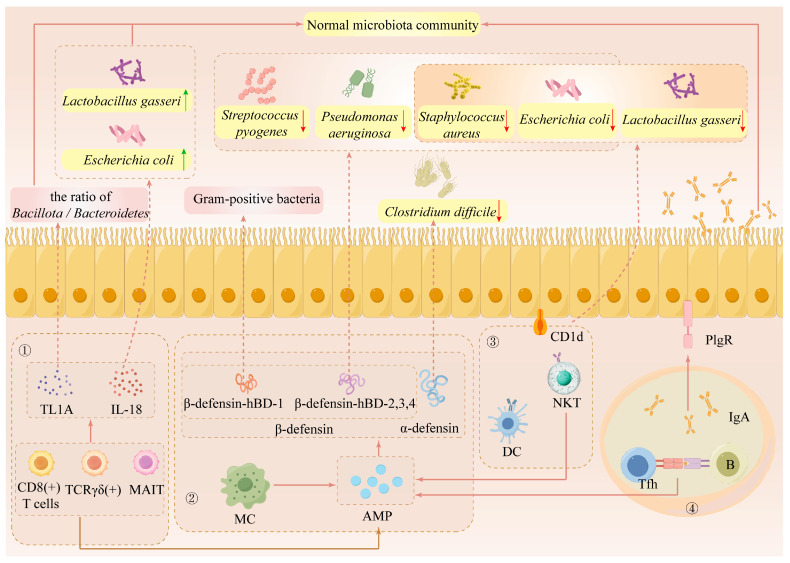
Inflammatory cells and inflammatory cytokines reshape the gut microbiota composition. The regulation of gut microbiota composition by host immune cells mainly includes ① direct regulation through inflammatory cytokines; ② regulation through antimicrobial peptides (AMP); ③ regulation through specific recognition of antigens; and ④ regulation through the production of specific antibodies.

**Table 1 foods-14-03836-t001:** Major cellular sources and physiological functions of inflammatory cytokines in the gut.

Inflammatory Cytokines		Major Sources	Main Physiological Functions	Ref.
Pro-inflammatory cytokines	TNF-α	T cells, macrophages, dendritic cells	Inflammatory; tumor necrosis; stimulation of adhesion molecules; activation of the immune system	[38]
	IL-12	Dendritic cells	Promote inflammation; promote the differentiation and proliferation of Th1 cells, enhance cellular immune response	[39]
	IL-17	Th17 cells, Treg cells, dendritic cells	Activate the signal cascade reaction; mediate the expression of inflammatory mediators in target cells	[40]
	IL-23	Dendritic cells	Reprogrammed immune cells; induce and maintain a highly pro-inflammatory state	[41]
	IFN-γ	T cells (Th1 cells)	Pro-inflammatory; promote Th1 immune response; activate macrophages	[42]
Anti-inflammatory cytokines	IL-10	Macrophages, dendritic cells, B cells	Anti-inflammatory; inhibit the release of Th1 cells and cytokine IFN-γ; inhibit the production of macrophage cytokines	[32]
	TGF-β	Dendritic cells	Immunity; regulating cell survival, metabolism, growth, proliferation, differentiation, adhesion, migration and death	[43]
Regulatory cytokines	IL-6	Dendritic cells, monocytes	Proinflammatory or anti-inflammatory; acute phase response; induction of humoral immune response	[44]
	IL-18	Regulatory T cells	Enhance the secretion of Th1 cytokine IFN-γ;	[45]
	IL-22	B cells, type 3 innate lymphocytes	Enhance mucosal barrier repair and maintenance;	[46]

**Table 2 foods-14-03836-t002:** Associations between diseases, gut microbiota alterations, and inflammatory cytokines.

Disease	Changes in the Gut Microbiota	Changes in Inflammatory Cytokines	Ref.
Inflammatory Bowel Disease, IBD	*Eubacterium rectale*, *Faecalibacterium prausnitzii*, *Roseburia intestinalis ↓* *Bacteroides fragilis ↑*	IL-6, TGF-β, IL-17, IL-22 ↑	[60]
Ulcerative colitis, UC	*Faecalibacterium prausnitzii*, *Prevotella*, *Peptostreptococcus ↓*	TNF-α, IL-1, IL-6, IL-9, IL-13, IL-33 ↑ TGF-β, IL-10, IL-37 ↓	[61,62]
Sepsis	*Staphylococcus*, *Streptococcus*, *Enterococcus*, *Haemophilus ↑* *Akkermansia*, *Ruminococcus ↓*	CRP, IL-6, IL-10, TNF-α ↑	[56]
Depression	*Bifidobacterium*, *Escherichia coli*, *Lactobacillus*, *Bacteroides ↓*	TNF-α, IL-1β, IL-6 ↑	[57]
Crohn’s disease	*Escherichia coli*, *Enterococcus* ↑ *Bifidobacteria*, *Lactobacillus* ↓	IL-1, IL-17, L-22, IL-33 ↑	[63]
Gastric cancer	*Bifidobacteria*, *Lactobacillus*, *Bacillus or Coccus* ↓	IL-6, IL-17 ↑	[64]
Non-infectious diarrhea	*Escherichia coli*, *Enterococcus* ↑ *Bifidobacterium*, *Lactobacillus* ↓	IL-2, IL-8, IL-10, TNF-α ↑	[65]
Alzheimer’s disease, AD	*Lactobacillus*, *Bifidobacterium*, *Ruminococcus* ↓ *Escherichia coli*, *Enterococcus* ↑	TNF-α, IL-6 ↑	[66]
Ankylosing spondylitis	*Cyanobacteria*, *Deinococcota*, *Patescibacteria*, *Actinobacteriota*, *Synergistota* ↑ *Acidobacteriota*, *Bdellovibrionota*, *Campylobacterota*, *Chloroflexi*, *Gemmatimonadota*, *Myxococcota*, *Nitrospirota*, *Proteobacteria*, *Verrucomicrobiota* ↓	IL-23, IL-17, IFN-γ ↑	[67]
Nonalcoholic steatohepatitis	*Bifidobacterium*, *Lactobacillus ↓* *Enterobacter*, *Enterococcus ↑*	IL-10, IL-17 ↑	[68]
Asthma	*Bifidobacteria and Lactobacilli ↓* *Escherichia coli*, *Helicobacter pylori*, *Streptococcus*, *and Staphylococcus aureus ↑*	CRP, TNF-α, IL-6 ↑	[69]

Note: “↑” indicates an increase in bacterial abundance or inflammatory cytokine levels; “↓” indicates the opposite.

**Table 3 foods-14-03836-t003:** Effect and mechanism of probiotics on inflammatory cytokines.

Experimental Methods	Year	Probiotic Strain	Diseases	Inflammatory Cytokines Changes	Improvements in Clinical Symptoms or Histological Indicators	Ref.
Animal experimentation	2025	*L. rhamnosus* GG	Inflammatory colitis	IL-10 ↑	Body weight ↑; Alleviated colon shortening; Histological scores ↑	[135]
Animal experimentation	2019	*B. infantis*	Inflammatory bowel disease, IBD	The expression of IL-10 and TGF-β1 ↑	Body weight ↑; Disease activity index (DAI) and histological damage scores ↓	[151]
Animal experimentation	2022	*L. rhamnosus* ZFM231	Colitis	TGF-β ↑; TNF-α ↓	DAI and colon tissue damage ↓; Body weight ↑;	[152]
Clinical trial	2020	*B. Longum* ES1	Irritable bowel syndrome, IBS	IL-6, IL-8 and TNF-α ↓	Key clinical symptoms and quality of life were significantly improved.	[153]
Animal experimentation	2021	*L. plantarum* HM-22	Be allergic	IL-10, IFN-γ, TGF-β ↑; Total IgE and IL-4 ↓	Body weight ↑; Abnormal organ indices and colon tissue damage ↓	[154]
Clinical trial	2022	*L. plantarum* IS-10506	Atopic dermatitis	IL-4 and IL-17 ↓; IFN-γ and Foxp3+ ↑	Scoring atopic dermatitis index ↓	[155]
In vitro experiment	2019	*L. salivarius* LS01 *L. acidophilus*	Parkinson’s disease, PD	TNF-α, IL-6, and IL-17A ↓; IL-4 and IL-10 ↑	Inflammatory cytokines and ROS ↓	[156]
Animal experimentation	2020	*L. plantarum* CRL 2130 *S. thermophilus* CRL 807 *S. thermophilus* CRL 808	Parkinson’s disease, PD	IL-6 and TNF-α ↓; IL-10 ↑	Effectively improved motor symptoms and neuroinflammation.	[157]
Animal experimentation	2021	*Lactobacillus plantarum* YS3	Ulcerative colitis, UC	IFN-γ, IL-1β, TNF-α, IL-6, IL-12 ↓; IL-10 ↑	DAI ↓; Alleviated colon shortening.	[158]

Note: “↑” indicates an increase in inflammatory cytokines levels or clinical symptoms; “↓” indicates the opposite.

## Data Availability

The original contributions presented in the study are included in the article; further inquiries can be directed to the corresponding authors.

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
