# Peer review of "Crosstalk Among Gut Microbiota, Microbial Metabolites, and Inflammatory Cytokines: Current Understanding and Future Directions"

_foods, 2025, doi:10.3390/foods14223836_

Round 1
Reviewer 1 Report
Comments and Suggestions for Authors
In this review, the authors discussed the interaction between the gut microbiota and the host immune system which can maintain health or contribute to disease. The authors elaborated the role of different microbial metabolites, such as short-chain fatty acids, tryptophan metabolites, bile acids, and polyamines in the modulation of the activity of immune cells and cytokines. The focus was also on the factors that affected the gut microbial composition and the levels of microbial metabolites that contributed to the regulation of inflammatory repose; these factors included antibiotics and probiotics. In short, this paper presented a comprehensive overview of relationships between gut microbiota, microbial metabolites, and inflammatory cytokines, and focused on how antibiotics/probiotics may alter that network of interactions. The authors mentioned that the finding presented in this paper suggest new therapeutic targets for immune-related diseases.
The paper is very well-written and the studies used in this review are up to date. The figures and tables presented are well prepared and informative. Minor revisions are needed as mentioned in the comments below.
-Line 28: “second organ” should be replaced by “second brain”
-Lines 58-59: Please add a reference at the end of the sentence.
-Lines 188-198: Please specify if the interleukins you are referring to is pro or anti-inflammatory such as IL-18 and IL-22 in this part.
-Lines 196-197: “. What’s more, butyrate also directly influences the plasticity of type 3 innate lymphoid cells (ILC3s), enhancing their production of IL-22 [93].”
Please remove “what’s more”.
-Lines 238-239: Please add a reference at the end of the sentence.
-Line 400: Please add a reference after the sentence ending with “pathological pathways”.
Reviewer 2 Report
Comments and Suggestions for Authors
The manuscript “Crosstalk among gut microbiota, microbial metabolites, and in- 2
flammatory cytokines: Current understanding and future directions” is an extensive review that analyze the interacions among the gut microbiota, microbial metabolites, and inflammatory cytokines, and explores the impact of antibiotics and probiotics on this interaction network. The manuscript is well structured and written, figures and tables are descriptive and well designed.
Some observation are:
- Some sections are hard to follow due to the use of many abreviations.
- Although the manuscript describes immune-related diseases, in some sections some pathogens are mentioned. In this regard, the associations between diseases, gut microbiota alterations, and inflammatory cytokines in infectious gastrointestinal diseases is not clear (common pathogens).
- Revise the references format.
Reviewer 3 Report
Comments and Suggestions for Authors
This review offers a comprehensive overview of the interplay between gut microbiota, their metabolites, and the host immune system. You have synthesized a vast amount of literature, especially on specific metabolites like SCFAs and tryptophan. However, I am concerned about its suitability for the Special Issue "Lactic Acid Bacteria in Functional Foods: Health Effects, Current Applications, and Future Trends". The manuscript primarily focuses on the general dialogue between the gut ecosystem and host immunity, with lactic acid bacteria discussed only as one component of the broader narrative. Additionally, the connection to "Functional Foods" is underdeveloped, lacking a deep exploration of incorporating viable LAB into foods, which is essential for this Special Issue. While the manuscript has strong points, its broad focus makes it more suitable for a general microbiology journal like Microorganisms.
My feedback for the authors is as follows:
• The manuscript effectively discusses the interactions between gut microbiota, microbial metabolites, and inflammatory cytokines. However, it should include a discussion on the NLRP3 inflammasome, which links microbial signals to host inflammatory responses. Gut microbiota components like lipopolysaccharides, ATP, and metabolites such as SCFAs can modulate NLRP3 activation in intestinal macrophages and epithelial cells. This activation triggers the release of IL-1β and IL-18, essential for gut immune balance but can lead to inflammation if dysregulated. Incorporating this aspect would enhance the manuscript's depth.
• The manuscript discusses interventions targeting gut microbiota. The authors describe three major types: probiotics, antibiotics, and engineered probiotics. While probiotics are emphasized as the primary intervention, it would be beneficial to expand this discussion. Including prebiotics, synbiotics, postbiotics, dietary modulation, fecal microbiota transplantation, and even phage therapy would provide complementary or alternative strategies for targeting gut microbiota-immune interactions. Incorporating these elements would enhance the review's translational scope and overall completeness.
• Section 3 provides a useful overview of microbial metabolites involved in cytokine regulation; however, it remains incomplete and conceptually inconsistent. The current classification of metabolites into dietary-derived, host-modified, and microbiota-derived categories is biochemically imprecise and partially overlapping. For instance, tryptophan metabolites are diet-origin but microbially synthesized, and bile acids are host-derived yet microbially modified—thus, neither fits neatly into a single category. Moreover, polyamines are not exclusively microbial products, and several key immunomodulatory metabolite classes are omitted, including lipopolysaccharides, lipoteichoic acid, bacterial extracellular vesicles, vitamins (B-group and K), peptidoglycan fragments, and phenolic derivatives. These compounds affect TLR, NOD, and inflammasome signaling pathways, influencing cytokines such as IL-1β, IL-6, TNF-α, and IL-10. A revised and more accurate classification that reflects these mechanistic distinctions would greatly improve this section's scientific rigor and completeness.
• Table 1 summarizes several cytokines involved in gut immune regulation. However, the grouping and descriptions make it difficult to distinguish between pro-inflammatory, anti-inflammatory, and immunoregulatory mediators. To enhance clarity, I recommend restructuring the table to clearly categorize cytokines as pro-inflammatory, anti-inflammatory, or regulatory. This approach would better reflect their complex and context-dependent roles in intestinal immunity.
• Figure 1: Inflammasome signaling is missing, though it’s a major cytokine activation route. The figure is overly dense, with too many labels and acronyms. The arrows do not clearly indicate direction, and in some cases, it is unclear whether the regulation is stimulatory or inhibitory. I strongly recommend simplifying and redesigning for clarity: Add inflammasome (NLRP3–caspase-1–IL-1β/IL-18) to complete the mechanistic map. Additionally, clarify the interaction flow, and ensure the legend defines all abbreviations.
• The term “kinesin” appears only once in the manuscript, within the caption of Figure 1, where it is mentioned without any accompanying explanation or context (“... indirectly through kinesin or goblet cells”). The use of kinesin in this context is scientifically unclear and potentially incorrect, since kinesins are intracellular motor proteins involved in microtubule-based vesicular transport rather than direct mediators of gut microbial or epithelial interactions. If the reference to kinesin is intentional, a brief mechanistic clarification should be added to explain its role in cytokine release or epithelial signaling. Otherwise, the term should be removed to avoid confusion.
• Table 3: Lacks outcome relevance. It shows cytokine shifts but not whether these correlated with improved clinical or histological markers.
Round 2
Reviewer 3 Report
Comments and Suggestions for Authors
No further comments.